



# Geographical distributions of mesospheric gravity wave activity before and after major sudden stratospheric warmings observed by Aura/MLS

Klemens Hocke[1,2], Jonas Hagen[1], Franziska Schranz[1], and Leonie Bernet[1,2]

[1]Institute of Applied Physics, University of Bern, Bern, Switzerland
[2]Oeschger Centre for Climate Change Research, University of Bern, Bern, Switzerland

**Correspondence:** K. Hocke (klemens.hocke@iap.unibe.ch)

**Abstract.** Observations of the global distribution of mesospheric gravity wave activity are rare. To our knowledge there exist only a few articles showing global maps of gravity wave potential energy in the mesosphere derived from observations of the instrument SABER (Sounding of the Atmosphere using Broadband Emission Radiometry) on NASA's satellite TIMED (Thermosphere Ionosphere Mesosphere Energetics Dynamics). In the present study, we find that the geopotential height (GPH)

measurements of the instrument MLS (Microwave Limb Sounder) on NASA's satellite Aura are sensitive to mesospheric gravity waves with horizontal wavelengths between 200 and 1500 km. We apply a data analysis which evaluates the standard deviation of horizontal GPH perturbations at a fixed pressure level and along the orbit of the sounding volume of Aura/MLS. The orographic waves from the Southern Andes in August serve as a test signal for the horizontal resolution and sensitivity of the method. We find enhanced gravity wave activity in the lower, middle, and upper mesosphere in a small region over the

Southern Andes. It seems that the horizontal resolution of the mesospheric gravity wave maps provided by Aura/MLS is higher than those of TIMED/SABER. We apply the method to estimate the global distributions of mesospheric gravity wave activity before and after the major sudden stratospheric warmings (SSWs) of January 21, 2006, January 24, 2009, and January 6, 2013 using 30 day intervals of Aura/MLS observations of GPH. It seems that the gravity wave activity in the lower mesosphere over the subtropical convection regions of the summer hemisphere are decreased after the SSW of January 21, 2006. The gravity

wave activity in the lower and middle mesosphere over middle and high latitudes (40°N to 70°N) of the winter hemisphere is decreased after the SSW of January 24, 2009. The major SSW of January 6, 2013 is preceded by enhanced mesospheric gravity wave activity over Eurasia at high latitudes (40°N to 60°N). This asymmetric gravity wave activity in the lower mesosphere is coincident with a long-lasting stay of the stratospheric polar vortex mainly in the Eurasian longitude sector before the SSW of January 6, 2013. In case of the SSW 2009 and SSW 2013, the gravity wave activity is enhanced at latitudes poleward of 70°N

in the lower and middle mesosphere after the SSWs.

## 1  Introduction

Atmospheric gravity waves transfer momentum through the atmosphere, and the breaking of gravity waves changes the circulation of the atmosphere (Alexander et al., 2010). On the other hand, the circulation and wind shears affect the propagation of





gravity waves by wind filtering processes (Fritts and Alexander, 2003; Fritts et al., 2016). Gravity waves can be generated by orography, convective activity, and jet streams. The present study deals with inertia-gravity waves (or low-frequency gravity waves) and medium-frequency gravity waves. Inertia-gravity waves have horizontal wavelengths from several hundreds of km to about 2000 km, and they have periods from several hours to 24 hours. Medium-frequency gravity waves have horizontal

wavelengths between 150 and 300 km, and they have periods of 1 to 3 hours. Numerical simulations by Vadas et al. (2015) showed that ocean surface waves can generate medium- and high-frequency atmospheric gravity waves propagating up to the thermosphere.

Eckermann and Preusse (1999) found a strong inertia-gravity wave in the stratosphere above Southern Andes with a vertical wavelength of 6-7 km and a horizontal wavelength of about 400 km. The Andes mountain waves are regularly detected in high-

resolution meteorological analyses during Austral winter (Shutts and Vosper, 2011). Hendricks et al. (2014) investigated the cause of the stratospheric gravity wave belt that extends eastward from the southern Andes to southeast of Australia. They found that this belt of enhanced gravity wave activity is a robust climatological feature due to nonorographic tropospheric gravity wave sources: spontaneous emission from jets in rapidly evolving baroclinic systems, frontogenesis, and convection. Inertia-gravity wave generation around the polar vortex in the stratosphere above the Syowa station (69°S, 39°E) was investigated by

Sato and Yoshiki (2008). They explained that the observed inertia-gravity waves are generated by a spontaneous adjustment around the geostrophically unbalanced polar night jet. The spontaneous generation of inertia-gravity waves above the regions of strong curvature of the tropospheric mid-latitude jet stream was simulated in the study of O'Sullivan and Dunkerton (1995). Williams et al. (2003) studied inertia-gravity waves in the large-scale flow of a rotating, two-layer annulus experiment. A review about inertia-gravity waves from atmospheric jets and fronts was provided by Plougonven and Zhang (2014). Kumar

et al. (2011) observed inertia-gravity waves generated from tropical cyclones, while Ki and Chun (2011) investigated inertia-gravity waves associated with tropospheric deep convection in Korea. A global morphology of gravity wave activity in the stratosphere by using the GPS occultation data of the GPS/MET experiment was derived by Tsuda et al. (2000).

John and Kumar (2012) presented for the first time the global gravity wave climatology from the stratosphere to the mesosphere observed by the satellite experiment TIMED/SABER. They averaged the gravity wave potential energy in the height

interval from 60 to 80 km. Enhanced gravity wave activity is obvious over high latitudes in the North Atlantic longitude sector in January and over the South Atlantic at high latitudes below and between the southern tips of South America and South Africa in August. The mesospheric gravity wave maps of John and Kumar (2012) do not show enhanced wave activity over subtropical deep convection zones of the summer hemisphere. This might be due to the selection of the colour table. The association of enhanced mesospheric wave activity with the Andes mountain waves in August is not so clear as we will show later

for observations of Aura/MLS. It is an interesting question how the Andes mountain waves propagate from the stratosphere to the mesosphere and beyond. Numerical simulations by Vadas and Becker (submitted 2019) and Vadas et al. (2018) indicate that gravity wave breaking at the stratopause over Andes lead to generation of secondary gravity waves which propagate to the mesosphere where they can dissipate and generate tertiary waves. The secondary and tertiary gravity waves have medium-and large-scales with horizontal wavelength of 350 to 2000 km. The horizontal resolution of Aura/MLS is sufficient to observe

these waves. The transition from orographic primary to secondary gravity waves was recently detected in TIMED/SABER



observations over the Southern Andes in winter by Liu et al. (2019). Proedrou and Hocke (2014) simulated the generation of a secondary large-scale gravity wave in a high-resolution climate model experiment. A traveling atmospheric disturbance is generated by a soil colour change in the Sahara. The concentric wave is centered in the Sahara, and its wave front reaches after 9 hours Brazil where the wave interacts with the water vapour convection zone over Brazil. As result, a secondary large-scale
wave is generated in the lower troposphere with concentric wave rings outgoing from Brazil.

In the past, the Microwave Limb Sounder on the satellite Aura (Aura/MLS) provided deep insight into the global distributions of high-frequency gravity waves (Jiang et al., 2004a, b, 2005). The retrieved high-frequency gravity waves have horizontal wavelengths less than 140 km and vertical wavelength of about 10 km. The high horizontal resolution is reached by using the level1 data of Aura/MLS, namely the brightness temperature of 20 microwave channels. Wu and Eckermann (2008)
presented advanced global and regional distributions of high-frequency gravity wave activity. In addition, they evaluated the high-resolution data of ECMWF meteorological reanalysis. In the conclusions, they suggested a further study which retrieves inertia-gravity waves from the level2 data of Aura/MLS. Hocke et al. (2016) followed this way and presented global distributions of inertia-gravity wave activity in the stratosphere and mesosphere based on temperature measurements of Aura/MLS. Meanwhile we noticed that the geopotential height measurements of Aura/MLS are more sensitive to mesospheric gravity
waves though we cannot explain the reason. Observations of global maps of mesospheric inertia-gravity waves are quite novel. We are only aware of a study by Ern et al. (2011) who presented global maps of gravity wave momentum flux at 70 km altitude retrieved from TIMED/SABER observations and a study by John and Kumar (2012) which we discussed in the paragraph before.

The second theme of the present study are major sudden stratospheric warmings (SSWs), and we like to investigate how
major SSWs influence the global distribution of mesospheric gravity wave activity. Pedatella et al. (2018) give a short historical overview of research on SSWs. Then they describe how SSWs can influence the dynamics, energetics, and composition of the whole atmosphere including the ionosphere. The phenomenon SSW has already been investigated and reviewed in the past, e.g., Matsuno (1971); Scherhag (1952); Schoeberl (1978). In case of major sudden stratospheric warmings, planetary wave breaking in the middle atmosphere leads to a reversal of the polar stratospheric vortex in the winter hemisphere. The SSWs
generally induce a displacement of the polar vortex or even a splitting of the vortex in two or more vortices. It is also observed that the polar vortex is often centered outside of the pole before an SSW because of strong activity of planetary waves with zonal wave number 1 before an SSW (Matthias et al., 2012).

Venkat Ratnam et al. (2004) observed enhanced gravity wave activity about 10 days before the major Southern Hemisphere stratospheric warming in late winter/spring of 2002. They noted that the stratospheric gravity waves did not occur inside the
vortex but at the edge and outside of the stratospheric polar vortex. Flury et al. (2010) analyzed temperature profiles over Switzerland during several SSWs in January and February 2008. They found enhanced stratospheric gravity waves when the polar vortex edge is shifted towards Switzerland. Using radio occultations from COSMIC (Constellation Observing System for Meteorology, Ionosphere and Climate) and CHAMP (Challenging Minisatellite Payload), Wang and Alexander (2009) found four SSW events in the Northern hemisphere during the winter 2007/2008. Stratospheric gravity wave activity was enhanced
at the polar vortex edge, and the timing of the enhancements was within a couple of days at the central dates of the SSWs in



the Northern hemisphere. Stratospheric gravity wave activity was low inside the vortex core. On the other hand, mesospheric gravity wave activity is enhanced at the pole and may lead to the so-called mesospheric cooling which is often associated to sudden stratospheric warmings. Zülicke et al. (2018) found a coupling between mesospheric coolings and major SSWs in numerical simulations and suggested that deep zonal mean easterlies at 60°N may act as a gravity-wave guide.

Hoffmann et al. (2007) analysed mesospheric wind observations obtained by meteor and MF radars at high latitudes. They found an increase of eastward wind in the mesosphere up to 30 days after the major SSW in January 2006. In addition, the mesospheric turbulent energy dissipation rate ( a measure of gravity wave activity) was enhanced up to about 30 days after the SSW. They suppose that this increase is connected to the decrease of stratospheric planetary wave activity (zonal wave number 1) after the SSW. Hoffmann et al. (2007) also reported about a mesospheric cooling shortly before and after the major SSW.

These research results show that the behaviour of mesospheric gravity waves before, during and after major SSWs is of high interest. The behaviour is certainly not fully observed and understood yet. Our study explains the analysis of the Aura/MLS data in section 2. The resolution and accuracy of the derived global maps of mesospheric gravity wave activity are investigated in section 3 by using the gravity wave activity above the Southern Andes as a test signal. Section 4 presents and discusses the global maps of mesospheric gravity wave activity before and after the major SSWs of January 2006, January 2009, and January

2013. Concluding remarks are given in section 5.

## 2    Data analysis

We apply a similar data analysis as in Hocke et al. (2016) who derived inertia-gravity wave activity from horizontal temperature perturbations observed by Aura/MLS. However, we found that the horizontal perturbations of geopotential height (GPH) are a better measure of mesospheric gravity wave activity. It seems that the temperature perturbations of Aura/MLS are much too

small in the mesosphere and there are patterns of decreased temperature perturbations at the South Atlantic Anomaly (SSA) which are not confirmed by TIMED/SABER and which also do not occur in the GPH perturbations of Aura/MLS. We do not know why the mesospheric temperature perturbations of Aura/MLS are less reliable than the GPH perturbations. Please note that the gravity wave theory of Hines (1960) describes gravity waves as periodic, linear perturbations in pressure, density, horizontal and vertical wind. The observed GPH perturbations at a fixed pressure level directly correspond to the pressure

perturbations at a fixed altitude which would be induced by a gravity wave. Hence, it is justified to use GPH perturbations as a measure of gravity waves. The temperature perturbation can be derived from the perturbations in density and pressure by using the equation of state for ideal gases.

In addition, we changed the averaging method of the global maps. Hocke et al. (2016) derived at first daily maps of global gravity wave activity, and later they averaged for example 30 daily maps for getting the monthly average of gravity wave

activity. In the present study, we are binning all observations of a month into the selected latitude-longitude grid yielding the monthly mean map of global gravity wave activity.

The data analysis evaluates the horizontal GPH fluctuations at a fixed pressure level and along the orbit of the sounding volume of the Aura satellite. The advantage of evaluating the horizontal fluctuations is that background features such as the





stratopause or the mesopause are not misinterpreted as gravity waves. Further, large-scale equatorial waves, tides or planetary waves with horizontal wavelengths of several thousand km will not be mistaken for gravity waves. In addition the vertical resolution of the gravity wave maps is of the order of 3-6 km. Such a resolution cannot be achieved by high pass-filtering the vertical oscillations of GPH profiles.

The level2 data of Aura/MLS consist of atmospheric vertical profiles with a spacing of 165 km (1.5° along the satellite orbit which is sun-synchronous with an inclination of 98° and a period of 98.8 min (Waters et al., 2006; Schwartz et al., 2008). In the present study, we evaluate the mean and the standard deviation of 5 consecutive GPH values separately at the pressure levels 0.1 hPa, 0.01 hPa and 0.0022 hPa (corresponding to altitudes of about 64 km, 78 km, and 86 km respectively). Before calculation of the standard deviation, the data is detrended in order to remove the effect of large-scale, horizontal background variations. We

selected 5 consecutive measurement points, since over a distance of $5 \times 165$ km a straight line fit of the horizontal background variation is a good approximation. The 5 points are collected within a time interval of 2 min. In case of more measurement points (e.g., 7) the straight line fit approximation becomes more invalid since the background atmosphere non-linearly varies over long distances. Further, the horizontal resolution of the gravity wave maps would be reduced. On the other hand, in case of three consecutive points the standard deviation and the mean are not well defined, and the measurement noise may dominate.

Hocke et al. (2016) estimated the response of the method by means of artificial sine waves (with different horizontal propagation directions) which are sampled with a spacing of 165 km along the sounding volume orbit. They found that the method includes some noise from high-frequency and medium-frequency gravity waves. Fortunately, the amplitudes of these waves in nature are smaller than the amplitudes of inertia-gravity waves so that the noise and aliasing problem should be not a serious problem for the data analysis. In summary, Hocke et al. (2016) found that the derived standard deviation is a good proxy for

the inertia-gravity waves with horizontal wavelengths from 200 km to 1500 km.

    Jiang et al. (2004a) explained in detail that the variable angle $\alpha$ between the line-of-sight and the wave fronts can lead to measurement geometry biases. Preusse et al. (2002) investigated the sensitivity of space-based measurements of stratospheric mountain waves to the viewing geometry. For the purpose and conclusions of the present study these biases are not so relevant since we are mainly interested in the order of magnitude and the rough geographic distribution of inertia-gravity wave activity.

The vertical resolution of the GPH profiles of Aura/MLS ranges from 3 km in the stratosphere to 6 km in the mesosphere (Schwartz et al., 2008). The present study utilizes Aura/MLS data of the version 4.2. The valid range of the GPH profiles is from 100 hPa to 0.001 hPa. Data are filtered by using status, quality, threshold, and convergence values as indicated by the Aura/MLS science team Livesey et al. (2015). GPH perturbations with vertical wavelengths of 6 km to 30 km are expected to have the strongest response. The approximated dispersion relation of inertia-gravity waves under inclusion of the Coriolis

frequency $f$ is (Fritts and Alexander, 2003):

$$\hat{\omega}^2 = N^2 \frac{k_h^2}{m^2} + f^2 \tag{1}$$

where $N$ is the buoyancy frequency, $\hat{\omega}$ is the intrinsic wave frequency, $k_h$ is the horizontal wave number, and m is the vertical wave number. For a constant buoyancy frequency $N = 2\pi/5\text{min}$ and $f$ for 45° latitude, the intrinsic gravity wave periods are of





the order of 2-12 hours for horizontal wavelengths of 200-1500 km and vertical wavelengths of 6-30 km which most efficiently contribute to the standard deviation of the horizontal GPH perturbations. In the following we use the term gravity wave instead of inertia-gravity wave.

The gravity wave maps have a spacing of 5° in latitude and longitude. For each grid point, we are binning all observations
(standard deviations) of the given time interval which are within ±5° in latitude and longitude of the grid point location.

## 3   Mesospheric gravity wave activity above Southern Andes in August

The stratospheric and mesospheric gravity wave activity above the Southern Andes is most intense during winter (Jiang et al., 2005; John and Kumar, 2012). A possible explanation is that orographic waves can propagate into the stratosphere since the zonal mean wind is eastward at tropospheric and stratospheric altitudes during winter. Further, it can be assumed that the
stratospheric polar vortex which is strong during winter acts as a gravity-wave guide (Zülicke et al., 2018). In addition the stratospheric polar vortex can be regarded as a source of gravity waves (Venkat Ratnam et al., 2004). Thus, we are binning the Aura/MLS data (standard deviation of GPH) of August 2012, August 2013 and August 2014 (solar maximum and Austral winter) into the latitude-longitude grid which has a spacing of 5° in latitude and longitude as already mentioned in section 2.

Figure 1 shows a clear increase of the GPH perturbations in the lower, middle and upper mesosphere over Southern Andes. As
discussed by Hendricks et al. (2014) for the stratosphere, there is a tendency for enhanced gravity wave activity in a belt along the vortex edge from the Southern Andes to Australia. The high resolution of the gravity wave map can be easily estimated by the sharp decrease of gravity wave activity westward of the Southern Andes. It is a new result that the strong increase of gravity wave activity over the Southern Andes is also observed in the upper mesosphere. This finding might be important for evaluation of the numerical simulation studies about secondary and tertiary gravity waves over Southern Andes (Vadas et al., 2018; Vadas
and Becker, submitted 2019). The gravity wave maps of John and Kumar (2012) derived from TIMED/SABER are much more diffuse in the mesosphere compared to the maps of Aura/MLS in Fig. 1. This is possibly related to the horizontal sampling of TIMED/SABER which is about 350 km while Aura/MLS has a horizontal sampling of 165 km for the level-2 atmospheric profiles.

Another new result of Fig. 1 is the enhanced gravity wave activity over the tropospheric deep convection zones in the tropics
and subtropics of the Northern summer hemisphere. To our knowledge this theoretically expected feature was not observed yet for the upper mesosphere. Jiang et al. (2004b) found that gravity waves propagate poleward in the stratosphere, away from the deep convection zones in the tropical troposphere. They explained this behaviour of gravity wave propagation by means of the background filtering effect.

## 4   Mesospheric gravity wave activity before and after major sudden stratospheric warmings

In this section, we present global maps of mesospheric gravity wave activity before and after the central dates of the major SSWs of January 21, 2006, January 24, 2009, and January 6, 2013. The central date of the major SSW is given by the reversal





of the daily-mean zonal-mean wind from eastward to westward at the 10 hPa level at 60°N. A list with the central dates of the major SSWs is presented by Butler et al. (2017).

It is said that each SSW tells a different story, and we will see that we get different results for the global distributions of mesospheric gravity wave activity of the three selected major SSWs. An important choice is the selection of the time interval

for binning of the observations. We selected time intervals of 30 days before and after the central dates of the major SSWs. Previous results by Hoffmann et al. (2007) and Matthias et al. (2012) show that the major SSW can be regarded as a transition from a phase of increased stratospheric planetary wave activity (zonal wave number 1) before the SSW to a phase of decreased stratospheric planetary wave activity after the SSW. In addition, Hoffmann et al. (2007) observed strong eastward winds and enhanced gravity wave activity in the high latitude mesosphere over about 30 days after the SSW. For a test, we also changed

the time interval from 30 days to 20 and 10 days but the results were not improved by a shorter time interval. Since the statistics of the gravity wave maps is better for 30 day intervals, we keep this time interval for averaging in the present study.

## 4.1 The major SSW of January 21, 2006

The major SSW of January 21, 2006 was forced by an eastward propagating upper tropospheric ridge above the North Atlantic which initiated a subtropical wave breaking in the middle stratosphere at 10 hPa (Coy et al., 2009). The SSW was unusually

strong and prolonged with a strong upper stratosphere lower mesosphere (USLM) vortex as observations by Hoffmann et al. (2007) and Manney et al. (2008) showed. The strong USLM vortex after the SSW was accompanied by an elevated stratopause and strong descent of CO in the high-latitude mesosphere (Siskind et al., 2007; Manney et al., 2008, 2009). As already mentioned the stratospheric planetary wave 1 was strong in the weeks before the SSW (Hoffmann et al., 2007). The long phases of the atmospheric states before and after the SSW are appropriate for the 30 day interval of binning of the Aura/MLS GPH

profiles.

Figure 2 shows the global mesospheric gravity wave activity where the standard deviation of the horizontal GPH perturbations are taken as a measure of wave activity as described in the section 2. In the lower mesosphere at 0.1 hPa (ca. 64 km), the gravity wave activity is enhanced before the SSW over the tropical and subtropical deep convection zones of the Southern summer hemisphere (Fig. 2a). After the SSW, these tropical and subtropical enhancements are less clear suggesting that the SSW

had an effect on the equatorial mesosphere (Fig. 2d). At high latitudes in the lower mesosphere of the Northern hemisphere the gravity wave activity maxima seem to shift from the Atlantic sector (before SSW) to the Eurasian sector (after SSW). The study of Manney et al. (2009) showed that the stratospheric polar vortex was shifted to the Atlantic sector before the SSW. In the middle and upper mesosphere (Fig. 2b,c,e,f) the global field of gravity wave activity is more diffuse and we cannot see a significant change of the distribution before and after SSW. It might be that the gravity wave activity in the middle mesosphere

is a bit decreased after the SSW, especially in the Southern hemisphere.

Figure 3 shows the zonal mean of the standard deviation of GPH. Here it is obvious that there was a decrease of gravity wave activity at about 20-30°S in the lower and middle mesosphere. This is a new result and it seems that the SSW of January 2006 had an influence on the mesospheric gravity wave activity in the summer hemisphere. There is a SSW composite study by Kodera (2006) who found that tropospheric convective activity is enhanced in the tropics of the summer hemisphere after the





meridional circulation change induced by the SSW. However, this finding does not fit to the decrease in mesospheric gravity wave activity over the deep convection zones after the SSW in Fig. 3.

It is useful to test the influence of intra-seasonal changes on our analysis method. For this test, we take the Northern hemispheric winter 2010/2011 which was without an SSW. Figure 4 shows the zonal means and we can say that the differences

between the blue and red lines are here due to intra-seasonal changes which are certainly not related to an SSW since the winter 2010/2011 had no SSW.

## 4.2   The major SSW of January 24, 2009

The major SSW of January 2009 was unusually strong and characterized by the splitting of the stratospheric polar vortex (Kim and Flatau, 2010). This is conform with the predominance of a planetary wave with zonal wavenumber 2 (wave 2). The

Eliassen-Palm (EP) flux of wave 2 was the strongest since the winter 1978/1979 (Harada et al., 2010). Upward propagation of wave packets from an upper tropospheric ridge over Alaska was found as the cause of the stratospheric polar vortex split (Harada et al., 2010). Pedatella et al. (2014) compared numerical simulations of the major 2009 SSW obtained by four whole atmosphere models. They found different results for planetary and tidal wave propagation in the mesosphere and lower thermosphere after the SSW. The differences are possibly due to different parameterizations of gravity wave drag in the models.

The models and also Aura/MLS show a mesospheric cooling almost coincident to the sudden stratospheric warming at 70°N to 80°N. In addition, there is evidence for an elevated stratopause after the SSW though the elevated stratopause is not as pronounced as in case of the January, 2006 SSW. Yamashita et al. (2010) investigated the stratospheric gravity wave activity before, during, and after the SSW of 2009 using high-resolution reanalysis data of the European Centre for Medium-range Weather Forecast (ECMWF). The magnitude and the occurrence of gravity waves correlated with the location and the strength

of the polar vortex that was strongly disturbed by planetary wave growth. They found that gravity wave activity (gravity wave potential energy density) at 1 hPa at middle and high latitudes was clearly decreased in the weeks after the SSW.

Figure 5 shows the global mesospheric gravity wave activity (standard deviation of the horizontal GPH perturbations) as measured by Aura/MLS in January 2009. In the lower mesosphere at 0.1 hPa, the gravity wave activity is larger before the SSW than after the SSW. This is valid for middle and high latitudes. A similar decrease of gravity wave activity after the SSW

was reported by Yamashita et al. (2010) for the stratopause region.

Again, one can see the enhanced gravity wave activity close to the deep convection zones in the Southern tropics before the SSW (Fig. 5a). After the SSW, Fig. 5d shows a zonal asymmetry of the gravity wave activity with larger values above Eastern Siberia. In the middle and upper mesosphere, gravity wave activity is less structured and more uniform.

The zonal mean gravity wave activity is shown as function of latitude in Fig. 6. Before the SSW, gravity wave activity is

enhanced at 40°N to 70°N in the lower and middle mesosphere. Poleward of 70°N, the situation is reversed since the red line (after SSW) has larger values than the blue line (before SSW). In the upper mesosphere, the differences of gravity wave activity before and after the SSW are marginal.





### 4.3 The major SSW of January 6, 2013

The major SSW of January 6, 2013 was investigated by Coy and Pawson (2015) and Liu and Zhang (2014). It was a vortex-splitting type SSW. Contrary to the SSW of January 24, 2009, the wave-1 EP flux and not the wave-2 EP flux was strong before the SSW. Particularly, from December 20 to 31, 2012 the stratospheric polar vortex was shifted to Eurasia. The vertical flux
of wave activity at the tropopause occurred mainly over the North Pacific prior to the SSW and a Pacific blocking was present in the troposphere at 300 hPa (Coy and Pawson, 2015). Liu and Zhang (2014) also reported about tropospheric ridges over the west coast of North America. These observations agree with Martius et al. (2009) who found that vortex-splitting SSWs are often induced by tropospheric Pacific blockings. The Pacific blocking also explains why the stratospheric polar vortex was shifted over Eurasia prior to the SSW of January, 2013. Nath et al. (2016) found that upward propagating wave packets from
North Pacific and Siberian Tundra were possibly responsible for the wave-2 vortex-splitting event. Further they supposed that the wave packets are reflected back downward to central Eurasia due to the strong negative wind shear of the upper stratospheric polar vortex. As a consequence, a deep trough over Eurasia with extreme cold weather occurred in mid-January 2013.

Figure 7 shows the mesospheric gravity wave activity averaged over 30-day intervals before and after the major SSW of January 6, 2013. In the lower mesosphere, the enhanced wave activity over the deep convection zones in the Southern
tropics are obvious in Fig. 7a,d. Further, it is obvious in Fig. 7a that the wave activity is enhanced over Eurasia at middle and high latitudes before the SSW. After the SSW, this enhancement over Eurasia disappeared in Fig. 7d. In the middle and upper mesosphere (Fig. 7b,c,e,f), the wave activity is less structured and more diffuse. However, one can still see a slight enhancement of wave activity over Eurasia before the SSW.

Figure 8 compares the zonal means of the gravity wave activity before and after the SSW in January 2013. The differences
are quite small (less than 15%). In the lower mesosphere at 40°N to 60°N the wave activity before the SSW is larger than after the SSW. At latitudes poleward of 75°N, the wave activity after the SSW is higher than before the SSW. This agrees with Figure 6 (SSW 2009). Also, Hoffmann et al. (2007) observed enhanced gravity wave activity after the SSW in Northern Norway. The observed mesospheric cooling over the North pole during some SSWs might be induced by enhanced mesospheric upwelling due to increased mesospheric gravity wave flux from below.

It is interesting to focus on the zonal asymmetry of the mesospheric gravity wave activity before the SSW of January 2013 as visible in Fig. 7a,d. As already mentioned above, the stratospheric polar vortex was shifted over Eurasia from December 20-31, 2012 during a phase of a dominant and stationary planetary wave-1. Figure 9a shows the potential vorticity (PV) in the stratosphere at 10 hPa. The PV data are provided by the ERA-Interim reanalysis (Dee et al., 2011) of the European Centre for Medium-range Weather Forecast (ECMWF). The enhanced PV values (reddish area) indicate the stratospheric polar vortex
which was shifted over Eurasia. We computed for the same time interval the average mesospheric gravity wave activity using the Aura/MLS observations of GPH. Figure 9b clearly shows enhanced gravity wave activity at mid-latitudes in the lower mesosphere over Eurasia. The mesospheric gravity waves are mainly along the equatorward edge of the stratospheric polar vortex. A similar relation between the vortex edge and stratospheric gravity wave activity was reported by Yamashita et al. (2010) and Venkat Ratnam et al. (2004). Now, we found this relation for the lower mesosphere before the SSW of January





2013. However, we did not find a zonal asymmetry of mesospheric gravity wave activity when we screened the time intervals around the SSW 2009 and SSW 2006. Possibly, one needs a long phase of a stationary planetary wave-1 in the stratosphere as it occurred in December 20-31, 2013.

In the middle and upper mesosphere (Fig. 9c,d) the zonal asymmetry is less pronounced than in the lower mesosphere. However, it is still visible that the gravity wave activity is enhanced over Eurasia. Generally, the wave amplitudes are increasing with height, and the gravity wave distribution becomes more uniform and diffuse in the upper mesosphere. In so far, it becomes more difficult to find correlations between tropospheric/stratospheric wave sources and gravity wave activity in the middle and upper mesosphere.

## 5 Summary

We found that the GPH measurements of Aura/MLS are more appropriate than the temperature measurements for estimation of mesospheric gravity wave activity. For unknown reasons, the temperature distribution of Aura/MLS is too smooth in the mesosphere with standard deviations smaller than $1.5°K$. For verification of the sensitivity of the GPH measurements of Aura/MLS to medium-scale gravity waves, we used the orographic waves over the Southern Andes during winter (August) as a test signal. Aura/MLS observed enhanced wave activity closely located to the Southern Andes in the lower, middle, and upper mesosphere. This hot spot of wave activity also gives a rough estimate of the horizontal resolution of the observation and analysis method which is roughly about $10°$ in latitude and longitude.

This positive result motivated us to investigate the behaviour of mesospheric gravity wave activity before and after the major SSWs of 2006, 2009, and 2013. In case of the vortex-splitting SSWs of 2009 and 2013, we found that the lower and middle mesospheric gravity wave activity at $40°N$ to $60°N$ was stronger before the SSW than after the SSW. This finding agrees with Yamashita et al. (2010) who found that the magnitude and the occurrence of stratospheric gravity waves at 1 hPa are correlated with the location and the strength of the polar vortex. The strong planetary wave activity before an SSW is usually accompanied by a strong and displaced polar vortex while planetary wave activity and the polar vortex are weak after the SSW.

At latitudes poleward of $75°N$, the mesospheric gravity wave activity after the SSW is higher than before the SSW. This finding is conform with observations by Hoffmann et al. (2007) who also reported about the occurrence of a mesospheric coolings in case of a major SSW. This mesospheric cooling could be induced by enhanced mesospheric upwelling due to enhanced gravity wave activity in the polar mesosphere.

In case of the SSW 2013, a zonal asymmetry of mesospheric gravity wave activity was observed before the SSW. Enhanced values occurred over Eurasia. Then we focused on the time interval December 20-31, 2012 when the stratospheric polar vortex was stationary over Eurasia. Indeed, we found enhanced mesospheric gravity wave activity over the equatorward edge of the stratospheric polar vortex. There are only a few observational reports about global and polar distributions of mesospheric gravity waves. It seems that the higher horizontal resolution of Aura/MLS compared to TIMED/SABER is an advantage for providing new insights into geographical distributions of mesospheric gravity wave activity.





The SSW of January 2006 was remarkable since the enhanced mesospheric gravity wave activity over the deep convection zones of the Southern tropics was decreased after the SSW. The reason of the decrease after the SSW is unclear since it is not conform with the observations of enhanced tropical tropospheric upwelling after SSWs (Kodera, 2006). Generally we found for all three SSWs, that enhanced mesospheric gravity wave activity occurred over the deep convection zones of the Southern

5  tropics of the summer hemisphere. This is expected since convectively-generated gravity waves propagate from the troposphere into the stratosphere (Jiang et al., 2004b). However, we think that the upward propagation of convectively-generated gravity waves from tropospheric deep convection zones into the mesosphere was not observed before the present study.

*Code availability.*  Programs are available from KH upon request.

*Data availability.*  The Aura/MLS level-2 data are available at the Aura Validation Data Center (AVDC). Reanalysis data are provided by

10  ECMWF.

*Author contributions.*  KH performed the data analysis. KH, JH, FS, and LB discussed and interpreted the results.

*Competing interests.*  We have no competing interests.

*Acknowledgements.*  We thank the Aura/MLS team for the excellent measurement data. We are grateful to ECMWF for the ERA-Interim reanalysis data. The Swiss National Science Foundation (SNSF grant 200021-165516) and the University of Bern funded the study.





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



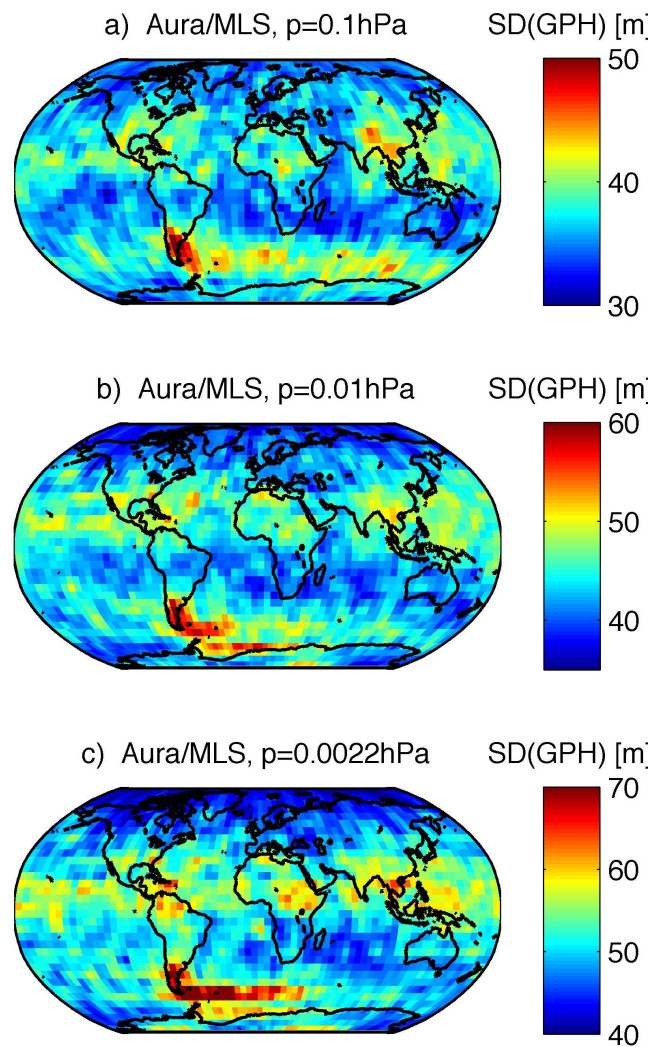

**Figure 1.** Mesospheric gravity wave activity estimated by the standard deviation of horizontal medium-scale fluctuations of geopotential height (GPH) observed by Aura/MLS in August 2012, August 2013, and August 2014. The maps are averaged for the pressure levels 0.1 hPa, 0.01 hPa, and 0.0022 hPa approximately corresponding to the altitudes 64 km, 78 km, and 86 km respectively.

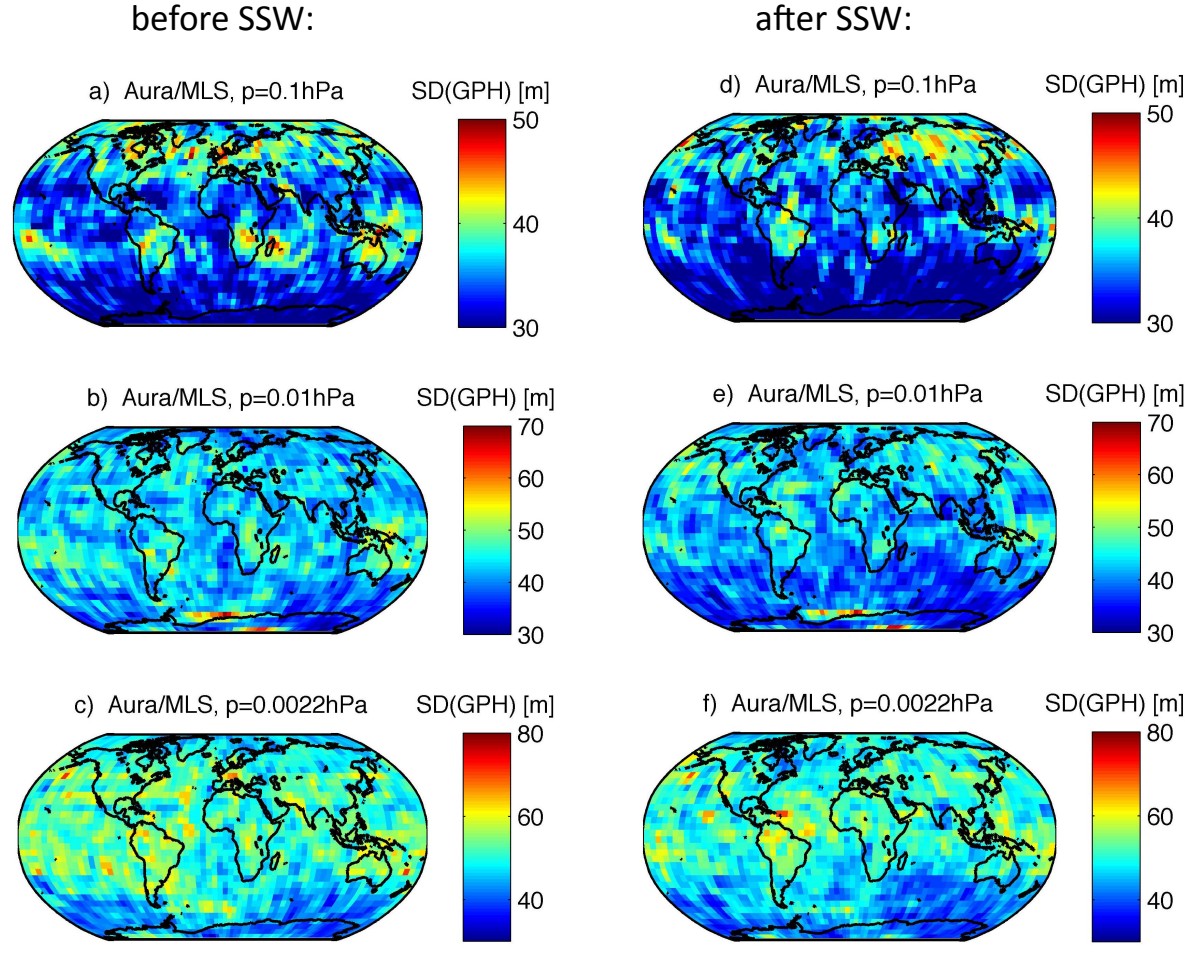

**Figure 2.** Mesospheric gravity wave activity estimated by the standard deviation of horizontal medium-scale fluctuations of geopotential height (GPH) observed by Aura/MLS before (a), b), c)) and after (d), e), f)) the major SSW of January 21, 2006 . The maps are averaged at the pressure levels 0.1 hPa, 0.01 hPa, and 0.0022 hPa for time intervals of 30 days before the SSW (a), b), c)) and 30 days after the SSW (d), e), f)).

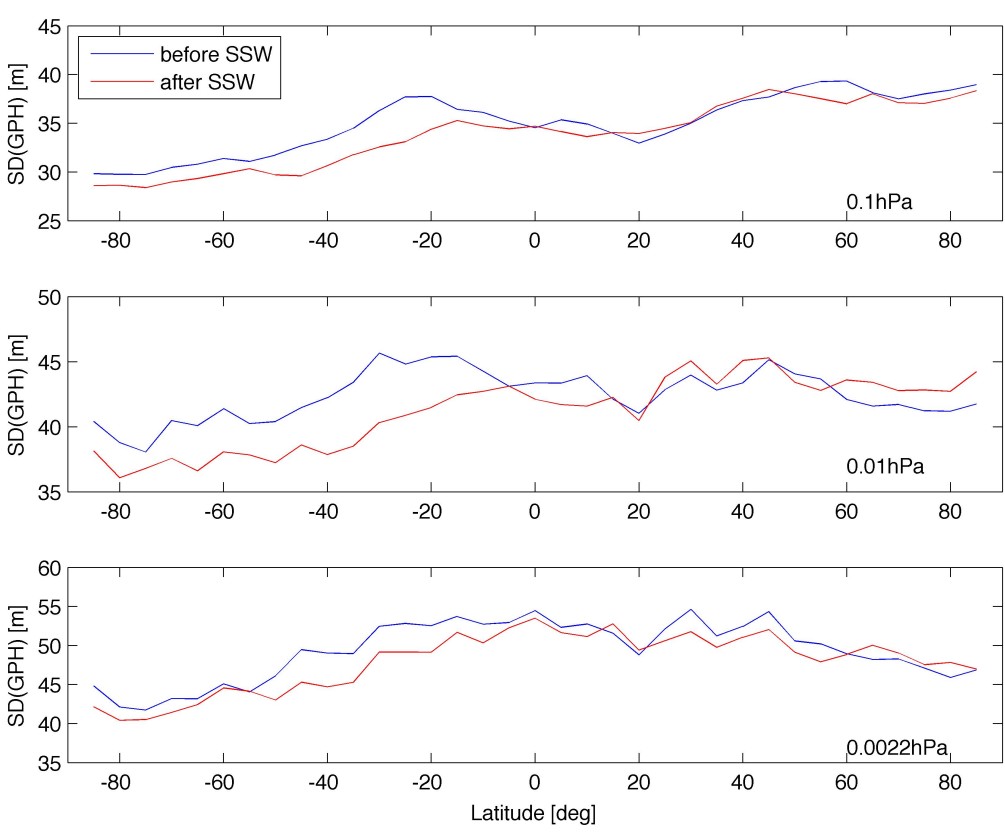

**Figure 3.** Zonal mean of mesospheric gravity wave activity estimated by the standard deviation of horizontal medium-scale fluctuations of geopotential height (GPH) before and after the major SSW of January 21, 2006. The diagrams are averaged at the pressure levels 0.1 hPa, 0.01 hPa, and 0.0022 hPa for time intervals of 30 days before the SSW (blue) and 30 days after the SSW (red).



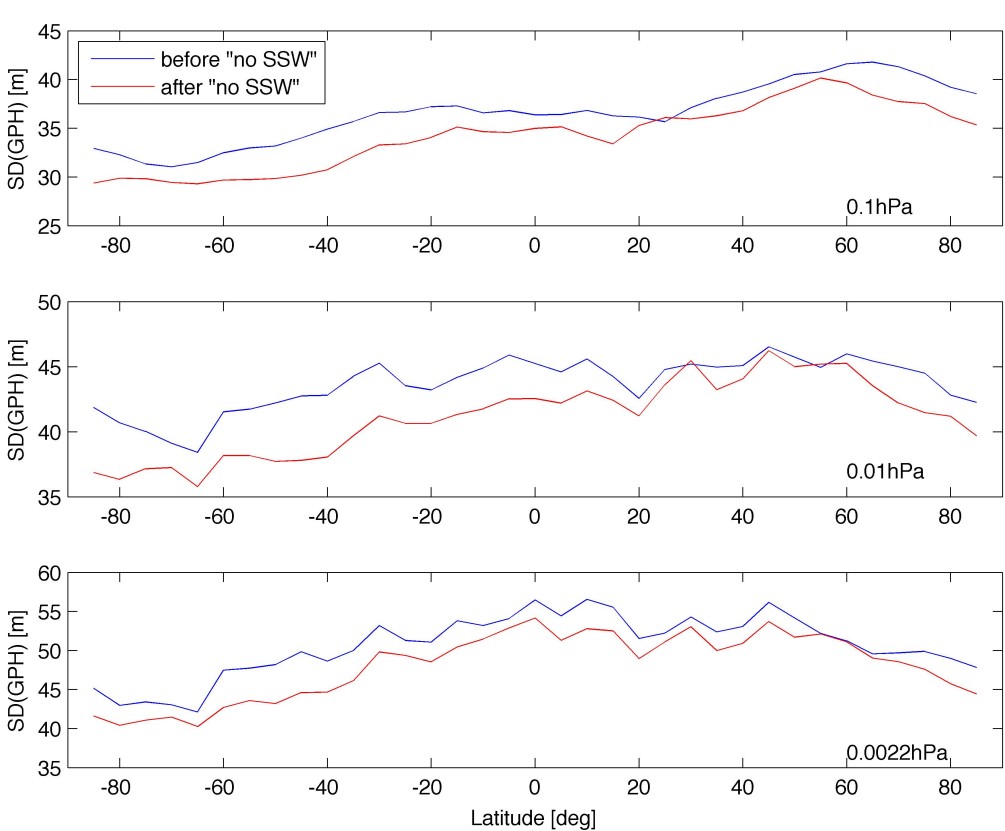

**Figure 4.** Same as Fig. 3 but for January 24, 2011 when no SSW occurred. This plot serves as a control case to estimate possible seasonal effects.



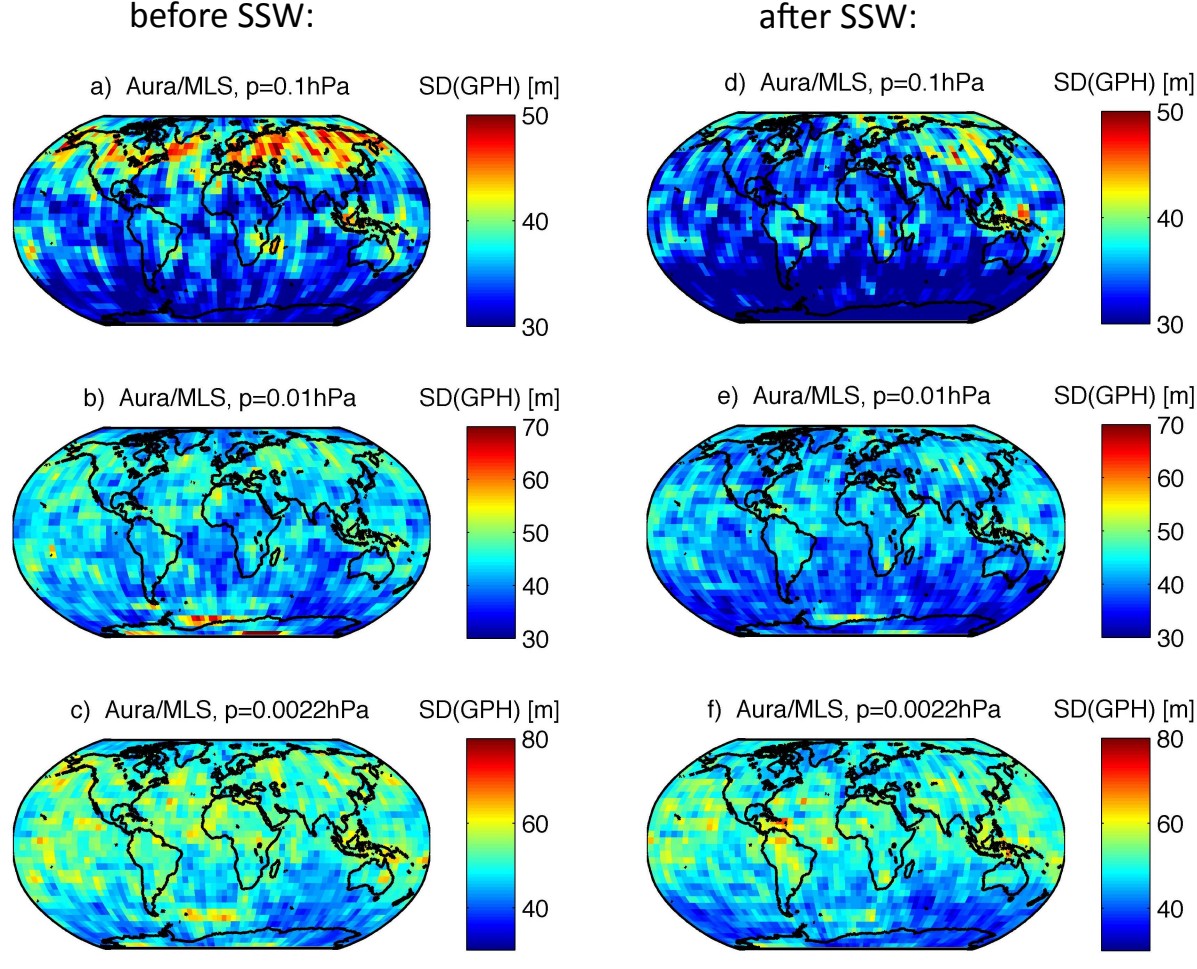

**Figure 5.** Mesospheric gravity wave activity estimated by the standard deviation of horizontal medium-scale fluctuations of geopotential height (GPH) observed by Aura/MLS before (a), b), c)) and after (d), e), f)) the major SSW of January 24, 2009 . The maps are averaged at the pressure levels 0.1 hPa, 0.01 hPa, and 0.0022 hPa for time intervals of 30 days before the SSW (a), b), c)) and 30 days after the SSW (d), e), f)).

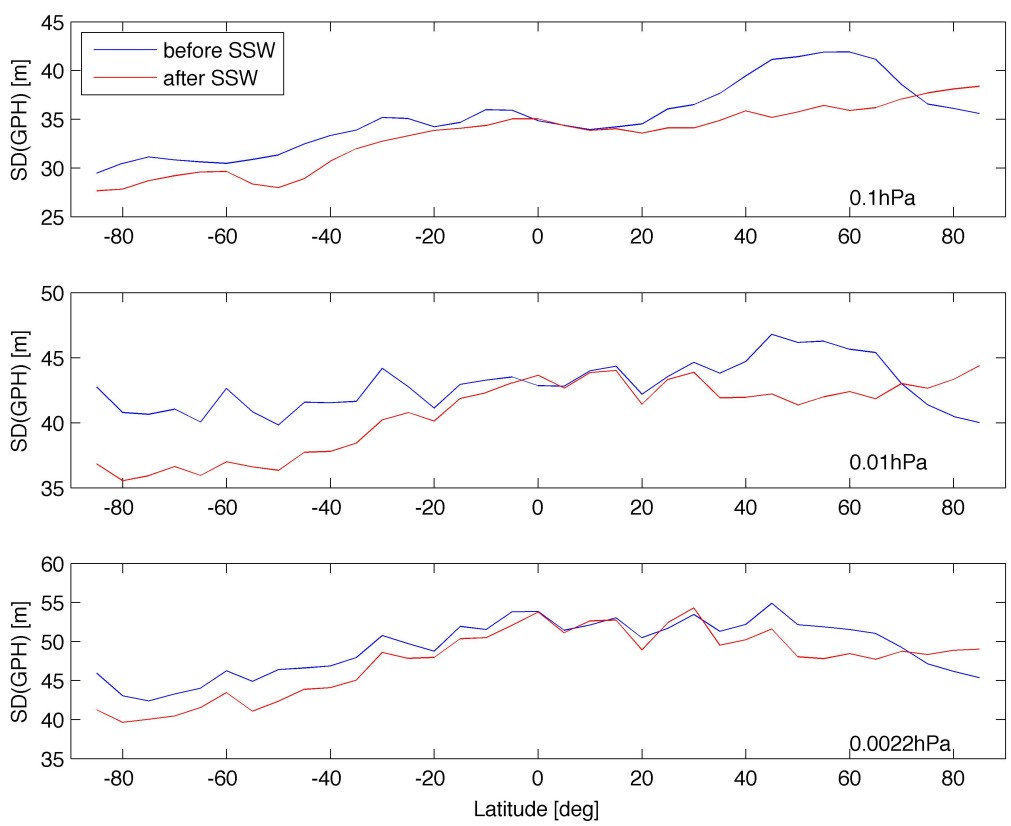

**Figure 6.** Zonal mean of mesospheric gravity wave activity estimated by the standard deviation of horizontal medium-scale fluctuations of geopotential height (GPH) before and after the major SSW of January 24, 2009. The diagrams are averaged at the pressure levels 0.1 hPa, 0.01 hPa, and 0.0022 hPa for time intervals of 30 days before the SSW (blue) and 30 days after the SSW (red).

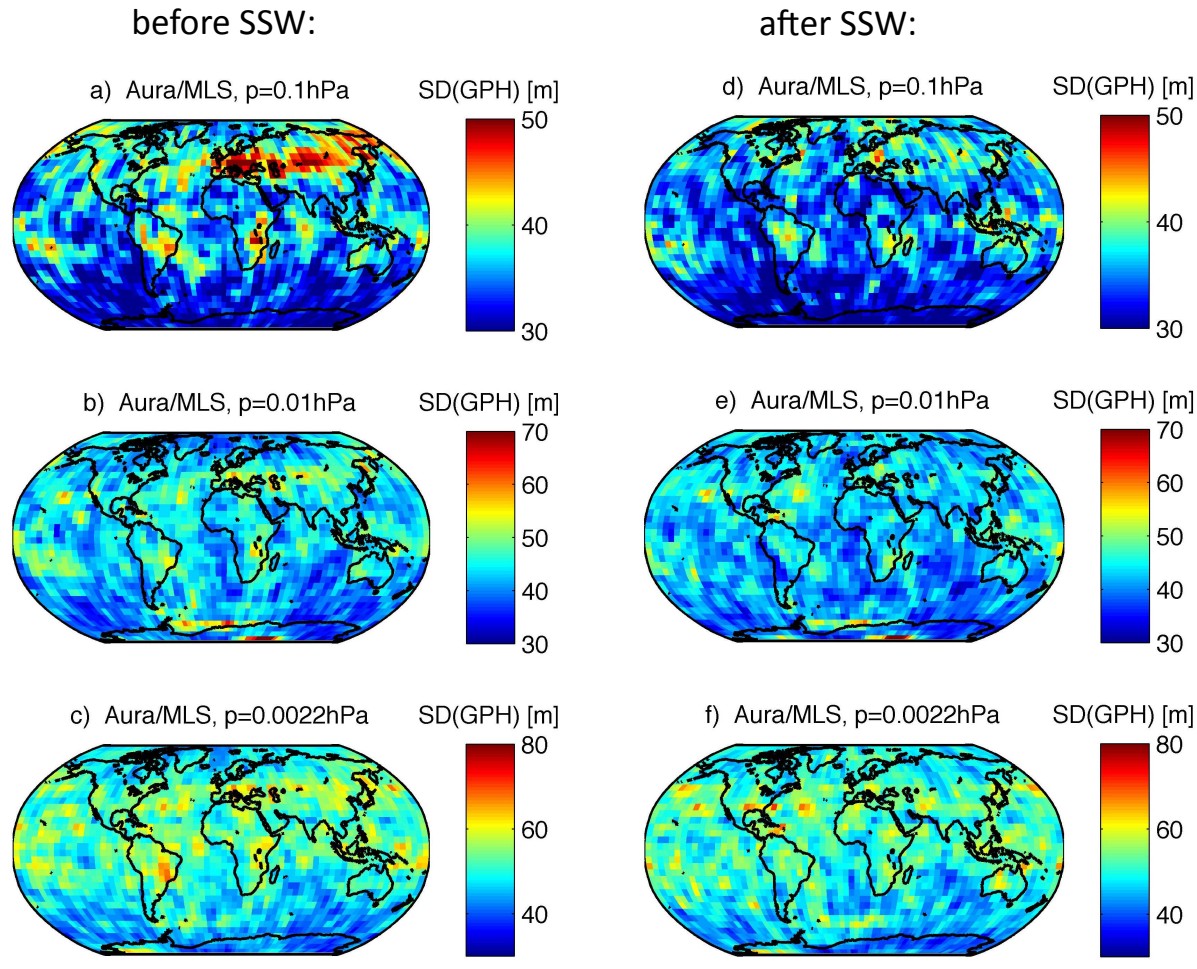

**Figure 7.** Mesospheric gravity wave activity estimated by the standard deviation of horizontal medium-scale fluctuations of geopotential height (GPH) observed by Aura/MLS before (a), b), c)) and after (d), e), f)) the major SSW of January 6, 2013 . The maps are averaged at the pressure levels 0.1 hPa, 0.01 hPa, and 0.0022 hPa for time intervals of 30 days before the SSW (a), b), c)) and 30 days after the SSW (d), e), f)).

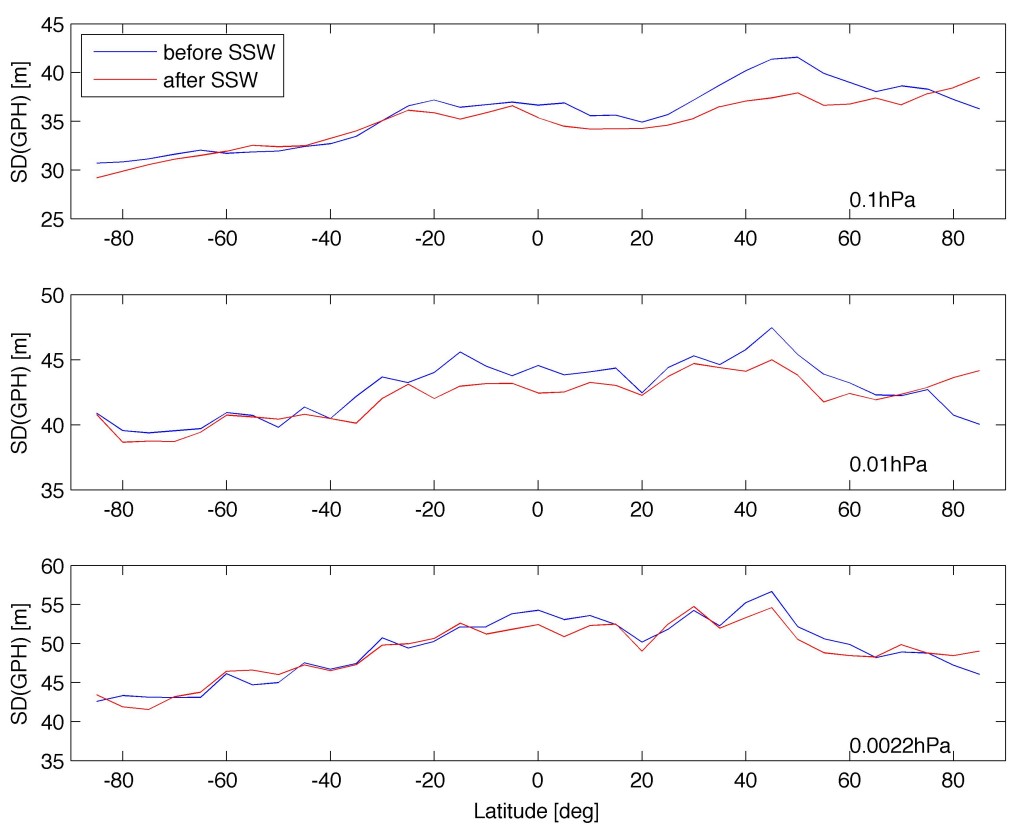

**Figure 8.** Zonal mean of mesospheric gravity wave activity estimated by the standard deviation of horizontal medium-scale fluctuations of geopotential height (GPH) before and after the major SSW of January 6, 2013. The diagrams are averaged at the pressure levels 0.1 hPa, 0.01 hPa, and 0.0022 hPa for time intervals of 30 days before the SSW (blue) and 30 days after the SSW (red).



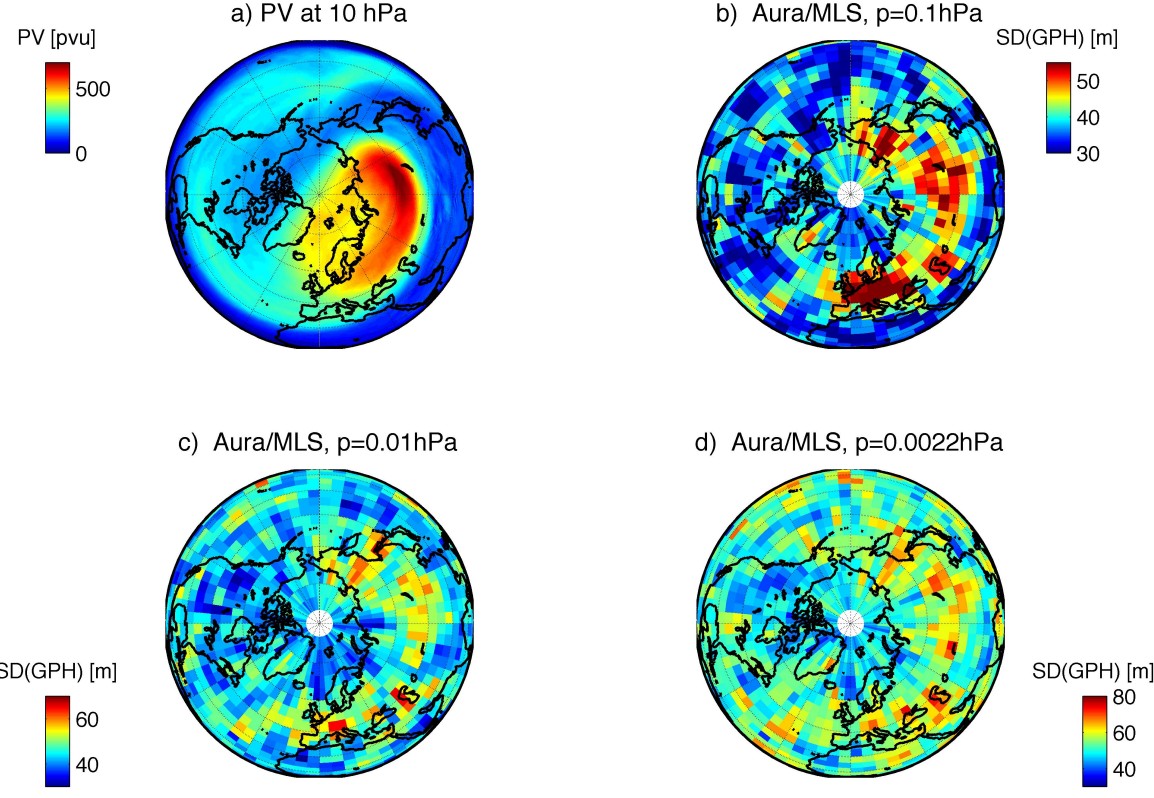

**Figure 9.** Polar plots averaged for the time interval December 20-31, 2012. a) ERA-Interim potential vorticity PV at the 10 hPa pressure level. [pvu] corresponds to $10^{-6}\,\mathrm{K\,m^2\,kg^{-1}\,s^{-1}}$. b) Lower mesospheric gravity wave activity at 0.1 hPa estimated by the standard deviation of horizontal medium-scale fluctuations of geopotential height (GPH) observed by Aura/MLS. c) Middle mesospheric gravity wave activity at 0.01 hPa. d) Upper mesospheric gravity wave activity at 0.0022 hPa.