# Peer review of "Geographical distributions of mesospheric gravity wave activity before and after major sudden stratospheric warmings observed by Aura/MLS"

_Atmospheric Chemistry and Physics, 2019_

## Referee Comment (RC1) · Anonymous Referee #1 · 23 Aug 2019

This manuscript, by Hocke et al, aims to measure and characterize the mesospheric distributions of gravity waves in the period immediately before and after sudden stratospheric warmings (SSWs). This is an important question. A range of previous studies have demonstrated that GWs are both affected by and potentially play a role in causing SSWs; furthermore, these waves can propagate significant distances to play an important role in stratopause and MLT dynamics.

Unfortunately, I am unable to support publication. This is primarily due to a fundamental methodological issue - without this issue I would recommend major revisions - but also

due to the structure of the study, which gives the impression of perhaps having been submitted prematurely. I am very sorry that the remainder of the review is so negative - I reiterate that this is an important and useful question, and encourage the authors to try again, with a different method and with more attention paid to contextualizing and generalizing their results.

The fundamental issue I refer to, which I believe undermines the results of the study, relates to the dataset used. The authors introduce a method of characterizing GWs based on the standard deviation of local along-track perturbations to geopotential height (GPH) as measured by the Microwave Limb Sounder (MLS) on NASA's Aura satellite. They do this because (page 4) "It seems that the temperature perturbations of Aura/MLS are much too small in the mesosphere" - this is an interesting conclusion, and previous work (e.g. Wright et al, AMT 2016, doi:10.1002/2013JD020526) supports it.

However, I do not believe that the **GPH** variable produced by MLS can be used in the way the authors do so. This is because of how MLS GPH is measured and calculated. Specifically, the MLS retrieval only directly measures GPH at the 100hPa altitude level (see Livesey et al 2006, IEEE Trans GRS, doi:10.1109/TGRS.2006.872327 - page 1147, right column). GPH values at all other altitudes are then computed by reference to the retrieved temperature profile, rather than directly from the observed radiances. Since the authors state that the temperatures are unsuitable for this analysis, I cannot see how their GPH-based results can be robust or meaningful. This is a critical issue, and therefore I am forced to recommend rejection for this reason alone.

However, in case I have misunderstood their method, the MLS GPH calculation method, or if they wish to repeat their analysis using different data and then resubmit, I include detailed further comments on the study below. If I am indeed incorrect in my understanding, then I think the paper could be considered acceptable with very major revisions, primarily of a textual and structural nature. In particular, an essentially complete rewrite of the Introduction would be almost required, and the SSW events

would have to be compared in a more meaningful and general way. I hope to see such a revised paper at some future date, as the results would interest me greatly.

Major Comments ===============

1. (critical) I do not believe the dataset they use can be used in this way. See detailed description above.

2. (critical) Even if I am incorrect about the validity of MLS GPH measurements for this purpose, the use of only five points to compute a standard deviation is deeply troubling. This is for two reasons. Firstly, the very small number of points is obviously a problem - a single outlier could skew the entire background fit. Secondly, the use of the standard deviations specifically introduces a parametric assumption that a normal distribution would accurately describe the variability of these points. Indeed, a range of previous studies (e.g. Hertzog et al JAS 2012, doi:10.1175/JAS-D-12-09.1) have demonstrated that GW phenomena tend to be log-Gaussian rather than true Gaussian in most variables, and I see no a priori reason to assume this is not the case for their GPH amplitudes. A better solution may be to first detrend the data, then combine them into the desired time-latitude-longitude-bins, and only then attempt to compute their statistical variability.

3. The paper seems to lack a central narrative - there seems to be no major attempt to generalize their results, or discuss how they, for example, differ systematically between split-vortex SSWs and displaced-vortex SSWs.

4. The Introduction is poorly written, and jumps from topic to topic. This makes it hard for the reader to draw a narrative through what the authors intend to investigate, and in particular makes it challenging to properly contexualize in the literature. Furthermore, many of the references are borderline-irrelevant. The most egregious example of this is the reference to Williams et al (2003). This is a study of a laboratory rotating annulus and a numerical simulation thereof, which could be of great relevance if the physics of that experiment were linked to the physics of this study by the authors. However,

no further context is provided about why it is relevant. This is not the only example of this, just a particularly jarring one. I strongly encourage a complete rewrite of the Introduction, maybe from a blank page.

5. The authors claim in several places to have shown things for the first time. This is usually not the case, and as a reviewer I tend to take such claims as an implicit challenge to disprove them. I provide some examples below in the Specific Comments section.

6. The measurements they make will have an angular projection bias, as described for example by Jiang et al (2004a), who they cite. The authors claim that this effect is minor for this study since they are considering only amplitudes. I do not believe this is a reasonable assumption. For example, consider a wave with phase fronts aligned at a small angle relative to the satellite scan track. While the wave itself may have very significant amplitude, the measured value by this method would be near-zero, since only a very small portion of the phase structure would be sampled. This is likely to introduce significant regional biases, especially between orographic and non-orographic wave regions, which have very different preferential wave orientations. While there's not much that can be done about it experimentally, the authors cannot just brush this issue away in a few words - it needs properly addressing at the very least as caveats in the appropriate places in the text, and preferably in discussion in the end as well.

7. The lack of time series (beyond just a simple division into before and after the SSW) majorly undermines the presentation of the results. It is well-known that GW properties vary significantly over the lifetime of an SSW - see for example Figure 3 of Wright et al 2010 (JGR, doi:10.1029/2009JD011858) - and a division in time this simplistic makes it hard to draw out possible mechanisms. This is primarily driven by the authors' choice of spatial binning resolution. MLS is a sun-synchronous limb sounder which orbits around the Earth $\sim$14 times per day. Thus, a one-day five-degree-squared bin will only contain a tiny number of points - single digit certainly, and very often (per-

haps modally) zero. This could be addressed by increasing the bin width to say 20-30 degrees longitude but retaining the latitude spacing - this would give enough points that meaningful time series could be generated, and the data could be considered as averages of much fewer than thirty days while retaining good signal to noise. Bins of this type are often used in limb sounder studies of GWs - see e.g. Alexander et al, JGR 2008, doi:10.1029/2007JD008807. In addition, there is no good reason that I can see to average over thirty days for their zonal-mean results - there should definitely be sufficient profiles available for meaningful values to be computed in the zonal mean at shorter timescales than thirty days.

8. The different SSWs and height levels are only compared by reference to descriptions of widely-separated figures. Some use of difference plots would be really useful to help explain the effects they see rather than trying to guide the reader to the right places across several panels. For example, Figures 3, 4, 6 and 8 are primarily discussed in terms of the differences between the red and blue lines - so why not just show these differences, preferably on a single combined panel so all the events can be compared to each other? Additionally, difference plots between the left and right columns of Figures 2, 5 and 7 would seem very useful for highlighting the geographic differences seen.

9. The color tables of Figures 1, 2, 5, 7 and 9 are hard to read, and introduce perceptual issues. Two examples - the yellow parts of the graphics will be strongly identified as a gradient-change feature by full-vision readers, while colorblind readers (around 8% of men, including several of my close colleagues) will be unable to tell between the red and green parts of the data. There is an extensive recent literature on this which I encourage the authors to consult for their future work, e.g. Hawkins, Nature 2015, doi:10.1038/519291d and references therein.

Specific Comments ==================

A: [P01 L01-03] The authors claim that only a few articles have plotted maps of mesospheric GW activity, and that all of these studies have used GWPE measurements from

SABER. This is not correct. To take a trivial example, the lead author of this paper is also lead author of a 2016 paper which shows maps of GW activity in the mesosphere derived from MLS temperature data using essentially the same method as this paper, which is cited in this study. Please clarify or rephrase.

B: [P01 L23] "On the other hand" to what?

C: [P03 L06] onwards: All of the described studies used the Microwave Limb Sounder which flew on UARS in the early 1990s, not the one which flew on Aura from 2004. They share a common design heritage, but are not the same instrument.

D: [P03 L25 onwards]: Some discussion of the differences in typical PW activity associated with splits and displacements would be useful here. For example, recent (i.e. post-2010) work by e.g. JG Esler, T Birner, A de la Camara and M Ern highlight that the interactions between GWs and SSWs are very different between split and displacement events, which may affect your results.

E: [P04 L20]: the anomalous temperatures over the SSA sound potentially very interesting - is this a well-known issue?

F: [P04 L21-24]: If the authors cannot work out why the GPH measurements are more reliable than temperatures (which I dispute above, on retrieval-method grounds), it would seem sensible to either try and work out why from first principles or to contact the instrument team and ask them for guidance. At the very least it would be helpful to quantify the issue.

G: [P04 L30 - P05 L01]: This is a fair assumption if the background features do not vary with height in the primarily-meridional scan track direction across a distance of $\sim$7 degrees (165x5km, at 112km per degree). Is this a reasonable assumption for the stratopause and mesopause?

H: [P05 L03]: The FWHM vertical resolution of the MLS temperature averaging kernels at mesospheric altitudes, from which the GPH data are derived, is roughly $\sim$6-8km (see

Figure 3.22.4 of the MLS v4.2 Data Quality Document, https://mls.jpl.nasa.gov/data/v4-2_data_quality_document.pdf). How then can they measure vertical features of order 3-6km?

I: [P05 L05]: MLS geolocation data (in the standard v4.2 HDF5 product, at least) are provided at a fixed height level (from memory, this is somewhere in the UTLS), but the actual profiles themselves slant with height due to the along-track travel of the satellite during measurement, in alternating close-wide pairs. This shouldn't affect their results except at very high latitudes where their data are very closely meridionally spaced, but should probably be clarified - the spacing is unlikely to be exactly 165km, but likely to be at two values spaced on either side of this value, and their five-point window will thus vary in size by roughly this value, which may be as much as a degree at these altitudes.

J: [Section 3]: Throughout the paper, the authors say they test their method over the Andes. However, what they actually do is plot global maps and then identify a peak over the Andes. This isn't a problem - they just need to be clearer about what they've done

K: [P06 L15]: Note that it is strongly disputed that this is the dominant mechanism for GW generation in this region. For example, observational and modeling studies (e.g. Sato et al JAS 2012 doi:10.5194/acp-15-7797-2015; Hindley et al ACP 2015 doi:10.5194/acp-15-7797-2015) suggest that a large fraction of these waves are related to refraction of Andean sources, while others (e.g. Alexander and Grimsdell 2013 JGR doi:10.1002/2013JD020526) suggest small islands are a key mechanism.

L: [P06 L18-19]: enhanced GW activity in the mesosphere over the southern Andes is in no way a new observation. See for example a vast range of radar and satellite studies in the mesosphere in this region, e.g. papers from the SAAMER radar authored DC Fritts and/or NJ Mitchell, which show high GW variances in this region which increase with height. This has also been shown in satellites, for example Figures 10 and 11 of

Wright et al (AMT 2016, doi:10.1002/2013JD020526) for SABER.

M: [Section 4.3] the authors show PV data for one warming event but not the others. Why is this? Also, ERA-Interim is now strongly deprecated and ECMWF recommend that these data not be used any more as the model underlying them is very old. I strongly encourage the authors to move to ERA5, which shows very significant improvements at high altitude relative to ERA-Interim when compared to observational metrics, and can be downloaded at a reduced resolution comparable to ERA-Interim making the dataset just as computationally tractable.

N: [Author contributions]: the authors do not specify the sources of several critical contributions. In particular, they do not say who wrote the actual text, or who plotted the results.

---

## Referee Comment (RC2) · Anonymous Referee #2 · 24 Aug 2019

General comments:

This study shows geographical maps of gravity wave activity in the mesosphere through standard deviations of geopotential height (GPH) obtained by Aura/MLS observation. This method is also applied for the investigation of the difference of gravity wave activity before and after three major sudden stratospheric warmings. Observational evidence of mesospheric gravity wave activity itself is quite interesting and is inherently important to understand the momentum budget in the middle atmosphere.

[Figure]

However, it seems that there are some problems for publication of this paper. One of the fundamental points is that the results are not carefully discussed and less convincing, in particular, in Section 4; for example, the differences of the gravity wave activities before/after SSW are not clearly separated by their intra-seasonal change despite their long average period – 30 days (albeit the authors try to discuss in Fig. 4). In other word, the SSW signals, which are considered as a change associated with SSW in this manuscript, seems to be mixed-up with the seasonal change shown in Fig. 4. On the other hand, the description of the methodology is not enough in Section 2 to convince readers that the method based on GPH perturbations would be reliable. In addition, in my knowledge, the upper stratospheric and mesospheric gravity wave activity averaged in August shown in Section 3 have been already examined by several previous studies. However, there is no appropriate reference, and this blurs novelties of the results using the high-resolution Aura/MLS data.

Overall, I cannot recommend this article for publication and would suggest re-submit with extensive revisions.

Major comments: 1. (Critial) The authors state that a decrease of gravity wave activity in the Southern summer hemisphere in the lower and middle mesosphere seems to be associated with the SSW of January 2006 in Section 4.1, and this is a new result. In my opinion, this discussion is not correct, since the decrease of the gravity wave activity in the Southern summer hemisphere is also seen in Fig. 4. This means that the decrease of the gravity wave activity is caused by the seasonal change of the background wind and/or of gravity wave sources such as convection/jet front. Because the authors use 30-day intervals for the statics, the effect of the SSW should be carefully distinguished from the seasonal change with "no SSW" years. For example, it is interesting that the gravity wave activity is enhanced at 0.01 hPa and 0.0022 hPa in the northern polar region after SSW in Fig. 3, which is not confirmed in Fig. 4. It is natural to think that this change is caused by the sudden disruption of the polar vortex. Thus, it may be better to define the effects of the SSW as anomalies from the seasonal change in no

SSW years. This point is related to Major comment 2.

2. (Critial) As mentioned in Major comment 1, it is important to elucidate the (intra-)seasonal change of gravity wave activity without SSW. In this manuscript, why do the authors test the seasonal change only in 2010/2011 winter? It would be better that such seasonal change should be constructed by averaging at least several years. Is this related to the limitation of the time-range in the used dataset? Please clarify that.

3. (Critial) In Section 2, the authors should explain the reason why the GPH perturbation is a better measure of gravity wave activities. At least, the comparison between the temperature perturbation and GPH perturbation of Aura/MLS should be shown as one figure. The reason why temperature perturbation is less reliable is described only based on the absence of the Southern Atlantic Anomaly, but this is not convincing. Please explain, or try to discuss this through the observational mechanism of Aura/MLS.

4. (Critial) About Section 3: In my knowledge, the enhanced gravity wave activity over and leeward of the Andes is not a new result. For example, Alexander (1998, JGR) showed maps of observed gravity wave radiance variances by the MLS observations at an altitude of 53 km, while Jiang et al. (2005, Advanced in Space research) also showed MLS normalized radiance variance map at 80 km. Walterscheid and Christensen (2016, JGR) showed maps of average wintertime standard deviations of the temperature profile over the altitude range 95-115 km using SABER temperature observation, while John and Kumar (2012, Clim. Dyn.) also showed global maps of gravity wave potential energies in the 60-80 km using SABER data. If the advantage of the methodology proposed by this study is high-resolution horizontal sampling, what is a new finding which has not been reported by the above previous studies? Or, is this just a confirmation of the obtained gravity wave activities by the proposed (new) method? Please clarify this point. The gravity wave activities likely caused by tropical convection have also been described by the observational studies, and the propagation paths are well examined by some modeling studies (for example, Sato et al., 2009,

JRL).

5. (Critial) It would be quite interesting to show that the horizontal maps of gravity wave activities along the edge of the polar vortex. However, the authors should pay careful attention to the reason why such an asymmetry cannot be found in 2006 and 2009 SSW. If the authors suggest that the stationary structure of the planetary waves, please show the time-series of the planetary wave structure.

6. Why the standard deviation of GPH perturbation shown in Fig. 1 does not significantly vary in the altitude direction? Does the GPH perturbation have similar meaning to the potential energy of gravity waves? If so, why doesn't the geographical distribution become diffusive in the upper mesosphere expected by the wave breaking mechanism?

Minor comment:

1. Page 2, line 4: why the periods of the inertia-gravity waves are limited to 24 hour? Due to the Doppler shift, the (ground-based) period can be longer than 24 hour.

2. Page 2, line 5: whey the periods of medium-frequency gravity waves are limited for 1 to 3 hours? Is there any derivation of the period range for gravity waves with horizontal wavelengths between 150 km and 300 km?

3. Page 2, line 12: Non-orographic gravity waves are not simply categorized as "tropospheric" gravity waves, since spontaneously generated gravity waves also originate from the upper part of the jet core (in the stratosphere)

4. Page 2, line 31: "...beyond, although recent modeling studies reveals propagation paths of gravity wave which focus into the polar night get (Sato et al., 2009, GRL; Shibuya and Sato, 2018, ACP)

5. Page 4, line 7: please delete a blank before "a measure"

6. Page 4, line 31: how frequently does the orbit of Aula/MLS cross a latitude-longitude

grid? I mean, how many times are data sampled per day at a latitude-longitude grid?

7. Page 5, line 5: Please add ")" at the end of the sentence.

8. Page 5, line 9: what do you mean by the "background"? Please clarify what has been done in the detrend process.

9. Page 5, line 24: In this sentence, the purpose of this study is to obtain the "rough" geographic distribution of gravity waves. This is controversial to the motivation using the high-resolution dataset.

10. Page 5, line 33: Is this value representative in the stratosphere or the mesosphere? And what is the season?

11. Page 6, line 23: I disagree with this sentence. I don't think that the horizontal sampling does have an impact to the horizontal map greatly. If so, please show 3 point-running averaged Aura/MLS map and confirm whether the map becomes diffusive as expected.

12. Page 5, line 27: The gravity wave propagation is not only filtered by the background wind, but is refracted by the gradient of the background wind.

13. Page 7, Line 28: To me, the difference between Fig 2a and 2d and that between Fig. 2b and 2e seems comparable.

14. Page 9, line 27: Why did the authors show the PV structure for 20-31 December (12 days), not for 30 days as the statics of the gravity waves? In addition, it would be better to use ERA5 reanalysis dataset and MERRA2 reanalysis dataset, since these datasets include the stratosphere up to 1 hPa, which may be suitable for the purpose of this study.

15. Page 10, line 6: This sentence is not convincing. One of the reasons why the gravity wave fields seems to be diffuse is because large-amplitude gravity waves are easy to collapse due to the wave breaking mechanism.

16. Page 11, line 2: "since it does not..."

---

## Author Comment (AC1) · 30 Oct 2019

We thank the reviewers and the editor for their efforts, comments and suggestions. We still think that the derived mesospheric gravity wave maps are reasonable. However, our future work will focus on a validation. Thus, we select the option of a resubmission of a changed study in some future, and we will not revise the present study.